# Three-dimensional-printed custom guides for bipolar coxofemoral osteochondral allograft in dogs

Christina C. De Armond[1], Stanley E. Kim[1]*, Daniel D. Lewis[1], Adam H. Biedryzcki[2], Scott A. Banks[3], James L. Cook[4], Justin D. Keister[3]

1 Small Animal Clinical Sciences, University of Florida College of Veterinary Medicine, Gainesville, Florida, United States of America, 2 Large Animal Clinical Sciences, University of Florida College of Veterinary Medicine, Gainesville, Florida, United States of America, 3 Department of Mechanical and Aerospace Engineering, University of Florida, Gainesville, Florida, United States of America, 4 Department of Orthopaedic Surgery, Thompson Laboratory for Regenerative Orthopaedics & Mizzou BioJoint® Center, University of Missouri, Columbia, Missouri, United States of America

☯ These authors contributed equally to this work.
* stankim@ufl.edu

**Data Availability Statement:** All relevant data are available from the Mendeley Data Repository at https://data.mendeley.com/datasets/2fb3r7dy9m/1.

## Abstract

The objective of this experimental study was to develop and evaluate a three-dimensionally printed custom surgical guide system for performing bipolar coxofemoral osteochondral allograft transplantation in dogs. Five cadaver dogs, weighing 20–38 kg were used in the study. Custom surgical guides were designed and three-dimensionally printed to facilitate accurate execution of a surgical plan for bipolar coxofemoral osteochondral allograft transplantation. Guide-assisted technique was compared to freehand technique in each cadaver. Surgical time was recorded and postoperative computed tomography and three-dimensional segmentation was performed. Femoral version and inclination angles, femoral neck length, and gap present at the femoral and acetabular donor-recipient interface was compared between the virtual surgical plan and postoperative outcome for both techniques. One-tailed paired t-test ($P < .05$) was used for statistical analysis. When compared to free-hand preparation, mean donor femoral preparation time was 10 minutes longer and mean recipient preparation time was 2 minutes longer when using guides (p = 0.011 and p = 0.001, respectively). No difference in acetabular preparation time was noted between groups. Gap volume at the acetabular and femoral donor-recipient interface was not different between groups. Mean difference between the planned and postoperative version angle was 6.2˚ lower for the guide group when compared to the freehand group (p = 0.025). Mean femoral neck length was 2 mm closer to the plan when using guides than when performing surgery freehand (p = 0.037). Accuracy for femoral angle of inclination was not different between groups. Custom surgical guides warrants consideration in developing bipolar coxofemoral osteochondral allograft transplantation as an alternative surgical technique for managing hip disorders in dogs.

**Funding:** Authors: SK, AB, SB. University of Florida College of Veterinary Medicine Fall Faculty Research Development Award Grant, 2017. The funders had no role in study design, data collection and analysis, decision to publish, or preparation of the manuscript.

**Competing interests:** The authors have declared that no competing interests exist.

## Introduction

Hyaline cartilage has long been recognized as a tissue with limited capacity for repair [1, 2]. Articular cartilage loss can precipitate whole-joint pathology with associated pain and dysfunction that may necessitate surgical intervention [3, 4]. Resurfacing of the joint with functional hyaline cartilage is therefore an attractive treatment strategy [5]. Implantation of osteochondral autograft plugs has been described in dogs with encouraging results but is associated with donor site morbidity and is usually limited to treating lesions $\leq 2$ cm$^2$. [6–8]. Osteochondral allograft transplantation procedures follow the same principles of osteochondral restoration as osteochondral autografting, but the graft is recovered from a qualified tissue donor of the same species. The use of an osteochondral allograft allows for resurfacing of larger defects using site-specific donor tissue and avoids donor-site morbidity [9–12]. Cartilage is immune-privileged such that donor hyaline cartilage with intact matrix is not associated with immune system rejection responses [13]. The subchondral bone component of an osteochondral allograft, however, is not immune-privileged and the recipient response must be dampened by limiting osteochondral allograft bone volume and removing donor marrow elements using pressurized lavage prior to implantation [14].

Large, multi-surface or bipolar (apposing surfaces) osteochondral allograft transplantation has been performed for human patients for more than 40 years [10–12]. Although success rates appear variable, reports suggest that use of osteochondral allografts with high chondrocyte viability at time of transplantation, autogenous bone marrow aspirate concentrate pretreatment of donor bone, accurate graft cutting and implantation techniques, and compliance with procedure-specific rehabilitation protocols can consistently result in highly successful 3- to 4-year outcomes [7, 8, 15–19].

One of the major technical challenges in effectively transplanting bipolar osteochondral allografts involves consistent, precise preparation of the donor graft and recipient bed. Maintenance of appropriate joint alignment is important for functional joint movement, and successful osseous integration by creeping substitution is facilitated by sufficient subchondral bone contact and graft stability [16–20]. Osteochondral allograft subchondral bone thickness between 3–8 mm has been advocated to confer structural integrity while minimizing recipient immune stimulation [14, 19]. Currently, grafts and recipient beds are manually prepared using common orthopedic instrumentation such as saws, reamers, burrs osteotomes, rasps, and rongeurs in a "freehand" technique [16, 19]. While this technique can be effective, the process is user-dependent, not standardized, and subject to human error.

Three-dimensional (3D) planning and custom surgical guide printing are being used with increasing frequency in human and veterinary orthopedic surgery [21–26]. The use of patient-specific osteotomy guides and instrumentation is associated with a high degree of accuracy and improved surgical efficiency [24, 27–29]. In dogs and cats, custom 3D printed guides have been used to facilitate limb deformity correction, fracture repair, and spinal surgery [22, 23, 25, 26]. Given the documented advantages of 3D virtual planning and 3D printed custom guides in other orthopedic applications, this technology may present similar benefits for performing bipolar osteochondral allograft transplantation. If this technology allowed for a standardized method for femoral head and acetabular osteochondral allograft transplantation that resulted in stable fixation, recapitulation of normal articular alignment, and successful osseous integration, biological hip joint restoration may become a suitable treatment option for the large number of dogs affected by disabling hip disorders [30].

The objective of this study was to develop a system of custom 3D-printed guides that would allow precise shaping of donor and recipient femoral heads and donor acetabula for bipolar osteochondral allograft transplantation for the coxofemoral joint in dogs. We hypothesized

that use of a custom guide system would result in faster osteochondral allograft and recipient site preparation, more precise osteochondral allograft-recipient fit, and improved accuracy in restoration of normal femoral head and neck version, inclination, and length compared to graft and recipient preparation performed without the use of guides.

## Materials and methods

### Subjects

This study was approved by the Institutional Animal Care and Use Committee (University of Florida IACUC Protocol #201710005). Ten mixed breed dog cadavers weighing 20–38 kg were used for this study. Pre-study power analyses were performed to determine the number of dogs needed to test our hypotheses with alpha = 0.05 and minimum power of 0.8. Using one-sample paired t-test size analysis, we determined that a minimum of 5 pairs of dogs was needed for testing our hypotheses. The dogs were skeletally mature, but age was otherwise unknown. There were seven intact male dogs and three female dogs who's reproductive status was unknown. The dogs were sourced from local animal shelters and had been euthanatized for purposes unrelated to this study.

### Preoperative imaging

Standard hips-extended ventrodorsal radiographs of the pelvis and both femurs were obtained to confirm the joints were appropriately sized and did not have advanced hip dysplasia or other coxofemoral pathology. The pelvis and pelvic limbs were then imaged using computed tomography (CT) (160 Slice Toshiba Aquillion CT Scanner, Cannon Medical Systems, Tustin, CA, USA). Helical volume data (slice thickness of 0.5 mm and 0.3 mm slice overlap) was acquired. The bone algorithm was used for all 3D reconstructions and analysis.

### Group allocation and 3D modeling

3D image processing and guide design were performed using 3D medical image processing software (Materialise Medical Imaging Software Suite, Materialise NV, Leuven, Belgium). Bone window volume Digital Imaging and Communications in Medicine (DICOM) files were imported into imaging software (Mimics, Materialise Medical Imaging Software Suite, Materialise NV, Leuven, Belgium) for segmentation and 3D image reconstruction of the individual femora and pelves. Stereolithography (.stl) files of the bones were then imported into 3D modeling software (3Matic, Materialise Medical Imaging Software Suite, Materialise NV, Leuven, Belgium) for development of the virtual operative plan and surgical guide design.

Once 3D images of each dog were created, acetabular diameter was used to match donor and recipient pairs. Acetabular diameter was measured on axial projections as the distance between the cranial and caudal acetabular border bilaterally for each dog. To ensure an appropriate fit, each dog's mean acetabular diameter (average length from cranial to caudal pillars, bilaterally) was used to pair donors with recipients, with an acceptable donor acetabulum diameter 1–4 mm smaller than the recipient acetabulum. Each hip (left or right) was randomly assigned in matched pairs to either the *guide group* or *freehand group*, where custom surgical guides were used during surgery for the guide group and the freehand group surgery was performed without guides.

### Femoral guide design

A biplanar osteotomy of the femoral head aligned along the physeal scar was planned to mimic the topography of the capital physis, simplify the osteotomy, and preserve the structural

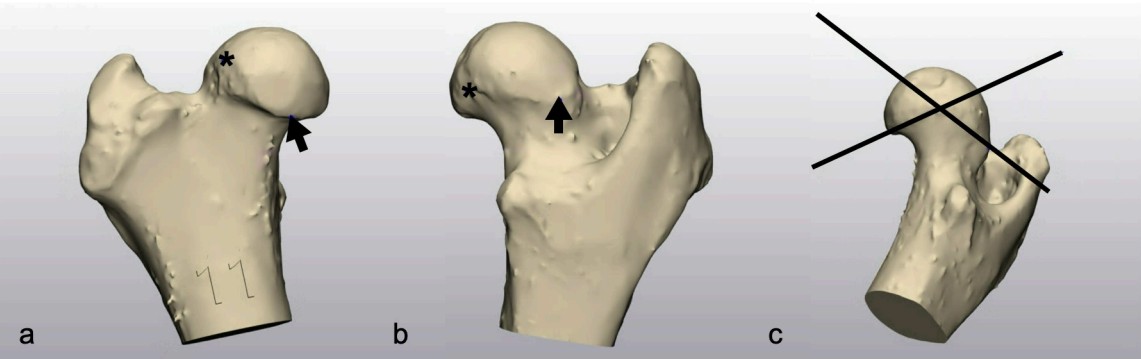

**Fig 1. Osteotomy plane creation.** Images show point selection on physeal scar and resultant cutting planes. (a) Craniomedial point is denoted by the arrow at the most distal aspect of the physeal scar on the cranial femur. Craniolateral point is marked by the asterisk at the most proximal border of the physeal scar, craniolaterally located on the femur. (b) Caudomedial point is shown by the asterisk at the proximomedial aspect of the physeal scar. Caudolateral point is marked by the arrow at the most distal point of the physeal scar, distolateral on the femoral neck. (c) Resultant osteotomy planes made from all four points. A craniodistal-to-caudoproximal plane was created using the craniomedial, caudomedial, and craniolateral points. A cranioproximal-to-caudodistal plane was created using the caudomedial, caudolateral, and craniolateral points.

integrity of the femoral neck (Fig 1). Virtual 3D models of both the donor and recipient femoral heads were superimposed by aligning four anatomic landmarks, or points, located along the physeal scar on each femoral head. The craniomedial point was located where the physeal scar curved distally on the craniomedial aspect of the scar (Fig 1a). The craniolateral point was located at the most proximal extent of the scar where the scar crossed over the proximal femoral neck (Fig 1a). The caudomedial point was located on the caudomedial aspect of the physeal scar, where the scar was most proximal (Fig 1b). The caudolateral point was located on the caudolateral side of the physeal scar, at the most distal point of the scar (Fig 1b).

Two planes were established based on the anatomic landmarks described above. A craniodistal-to-caudoproximal plane was created using the craniomedial, caudomedial, and craniolateral points. A cranioproximal-to-caudodistal plane was created using the caudomedial, caudolateral, and craniolateral points. (Fig 1c). These planes served as planned common osteotomies in a chevron configuration that would be performed on the donor and recipient specimens.

A donor femoral osteotomy guide was designed to conform to the topography of the donor femoral head and neck, and featured osteotomy shelves to guide sagittal saw placement when performing the osteotomies (Fig 2a). The shelves were positioned such that the subchondral thickness was < 8 mm in resultant ostectomized femoral head segment.

A recipient femoral osteotomy guide was designed to conform to the topography of the femoral head and proximal femoral neck. The guide included osteotomy shelves that would accommodate a sagittal saw blade. The osteotomy shelves were designed using the same planes as the donor guide, resulting in a complementary recipient bed for the prepared donor graft (Fig 2b).

## Acetabular guide design

Commercially available acetabular reamers (BioMedtrix, Whippany, New Jersey, United States) were used to prepare the recipient acetabular site in the manner of artificial total hip replacement [31, 32]. In order to match the hemispherical osteotomy created by acetabular reamers, a donor acetabular cutting jig was designed to be used in conjunction with a

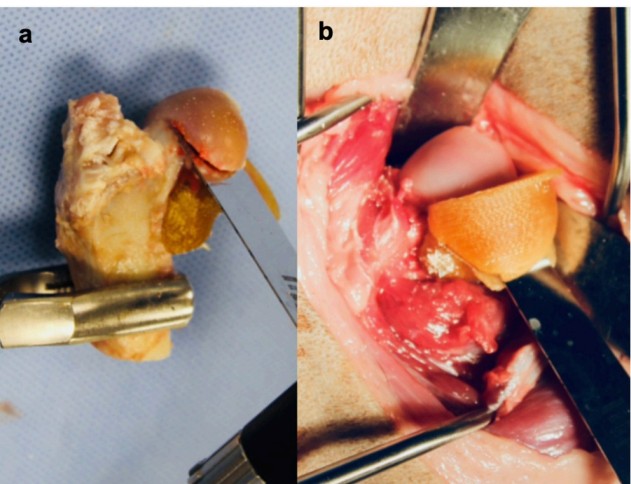

**Fig 2. Donor and recipient femoral osteotomy guides.** (a) Image shows the donor femoral osteotomy guide. A custom guide has been fit to the donor proximal femur with osteotomy shelves made from the osteotomy planes (Fig 1c). (b) Image shows the recipient femoral osteotomy guide. A custom guide has been fit to the recipient femoral head. Osteotomy shelves have been made from the osteotomy planes (Fig 1c).

pneumatic surgical drill (Hall 5058–01 Surgairtome Two, ConMed, Utica, New York, United States) and burr to sequentially contour the subchondral bone (Fig 3). The articular surface of the donor acetabulum was secured to a central post in the center of the base of the jig. The topography of the convex surface of the central post conformed to the articular lunate surface and acetabular fossa. A circular guide rail (Fig 3a) was designed to guide a burr during shaping of the donor acetabulum. The guide rail was secured to the base of the jig and could be rotated 360˚ around the central post that held the donor acetabulum. A burr support (Fig 3a) was designed to allow the drill and burr to move along the circular guide rail. The burr support

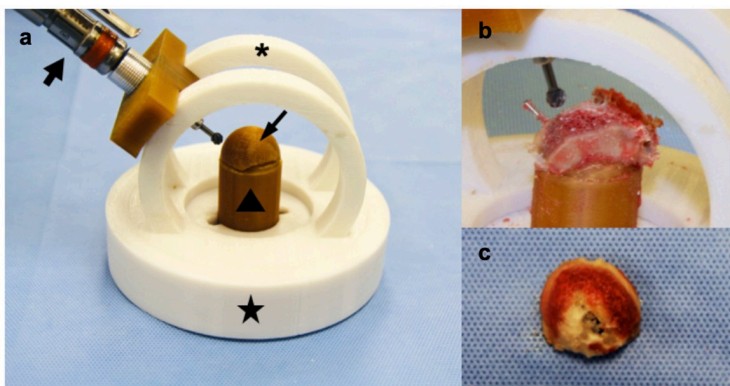

**Fig 3. Printed donor acetabular cutting jig and donor acetabular preparation.** (a) The circular guide rail (arc) (asterisk) can rotate freely within base platform (star). The donor acetabulum is secured to a central post (triangle). A 3-printed version of the virtually planned graft (check acetabulum) (thin arrow) is placed on the central tower and allows selection of burr height. The pneumatic burr (thick arrow) fits into the bur support with adjustable set-screw. By sliding the burr back and forth over the arc and rotating the arc around the base, a hemispherical subchondral bone surface is created. The check acetabulum is placed on the central post to determine drill height and adjust set-screw. (b) The donor acetabular segment is secured to the central post and the burr and arc are used to finish the subchondral surface shape. (c) Finished donor acetabular osteochondral allograft subchondral surface.

contained a set-screw that allowed adjustment of burr height in relation to the central post, thereby adjusting the graft thickness. A 3D-printed version of the virtually planned donor acetabular graft, or *check acetabulum template*, was used to determine the appropriate height of the burr (Fig 3a). By moving the burr back and forth along the guide rail and rotating the guide around the base of the cutting jig, a hemispherical donor acetabular graft of appropriate subchondral bone thickness could be fashioned.

All guides and bone models were 3D printed using a fused deposition modeler additive manufacturing printer (Fortus 450mc, Stratasys, Eden Prairie, Minnesota, United States). Biocompatible plastics used in printing included polyetherimide (Ultem™ 1010 resin, Stratasys, Eden Prairie, Minnesota, United States), polycarbonate ISO (PC-Iso, Stratasys, Eden Prairie, Minnesota, United States), and acetyl-butyl-styrene (ABS M30i, Stratasys, Eden Prairie, Minnesota, United States). The plastic used for each guide was dictated by availability of material at the time of printing.

## Surgery

All surgical procedures, including osteochondral allograft preparation and implantation, were performed by the same board-certified small animal surgeon (*SEK*) to limit variation in technique. Recipient femoral site preparation time recorded, which was the duration from recipient femoral head and neck exposure until completion of the osteotomies. Time to prepare each femoral and acetabular osteochondral allograft and the recipient femoral site was also recorded. Recipient acetabular site preparation time was not included for comparison as the same commercially available total hip replacement acetabular reaming instrumentation and technique were used for both groups [31, 32]. Implantation and fixation times were not included for comparison as the same technique was used for both groups.

**Guide group procedure.**   Cadavers were placed in lateral recumbency and a craniodorsal approach to the coxofemoral joint was made [33]. The coxofemoral joint was luxated and the recipient femoral guide was applied to the femoral neck and secured using a 0.9 mm Kirschner wires (Fig 2) A sagittal saw was used to create biplanar osteotomies with the guide shelves. The acetabulum was reamed in accordance with manufacturer recommendations for total hip replacement [31] after femoral head resection.

The osteochondral allograft femoral osteotomy guide was applied to the femoral neck and calcar region of the donor femur and secured using 0.9 mm Kirschner wires (Fig 2). Biplanar converging osteotomies were made along the osteotomy shelves to create the femoral osteochondral allograft (Fig 1c). The printed donor acetabulum template was applied to the acetabular guide central post and maximum burr advancement was determined (Fig 3). The printed donor acetabulum template was removed, and the harvested donor acetabulum was secured to the guide central post and surrounding bone was secured to the base using 1.1 mm Kirschner wires (Fig 3). Several broad cuts were made using a sagittal saw, and then the arc and burr were used to finish contouring the spherical subchondral surface (Fig 3).

The osteochondral allografts were secured in the recipient sites using Kirschner wires (Fig 4). Acetabular osteochondral allografts were secured with two wires: one at the level of the caudal pillar and the other at a more craniodorsal location (Fig 4) The femoral osteochondral allograft was secured to the recipient using parallel Kirschner wires aiming from the fovea capitis distally down the center of the femoral neck (Fig 4). The coxofemoral joint was reduced. No additional procedures were done to maintain the reduction other than routine closure of the joint capsule, tendon, fascia, subcutaneous tissues and skin.

**Free-hand procedure.**   After approaching the contralateral coxofemoral joint [33] and luxating the femoral head in the same manner as the guide group, biplanar osteotomies were

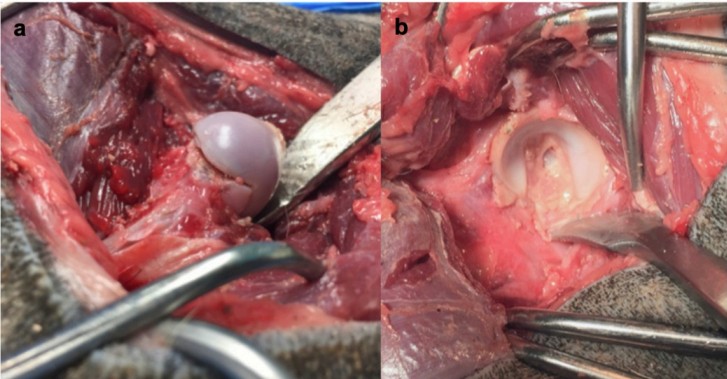

**Fig 4. Osteochondral allografts secured in recipient.** (a) Femoral osteochondral allograft attached to recipient femoral neck. (b) Acetabular osteochondral allograft implanted into recipient acetabular bed.

made in the femoral head at the level of the physeal scar using a sagittal saw to excise the articular surface. The osteotomies were made referencing the same landmarks outlined in the guide group but were performed without the assistance of guides. The acetabulum was reamed in standard fashion [31]. Donor osteochondral allografts were manually shaped using burrs, rongeurs, and a sagittal saw. The biplanar donor femoral head and neck osteotomy configuration was based on the four landmarks around the physeal scar as described for the guide-group; however, the osteotomies were performed without the assistance of guides. Donor acetabular preparation was performed freehand with the same spherical subchondral bone shape objective as the guide group. This was accomplished by first making broad cuts with a sagittal saw to remove the medial acetabular cortex, ilium, and ischium, exposing the acetabular subchondral bone. Progressively smaller sections of bone were removed using the saw. Rongeurs were used to remove prominent edges and smooth out a spherical convex subchondral surface. Final maximal subchondral bone thickness was measured to ensure it did not exceed 8 mm. Osteochondral allografts were secured in the same manner as done in the guide group coxofemoral joints (Fig 4) and closure was routine.

### Postoperative imaging and assessment

Postoperative pelvic CT scans were acquired using the same methods as described for preoperative planning. The size of the gap at the donor-recipient interface and femoral head alignment were calculated from the CT images. Acetabular alignment was not measured as the recipient acetabular bed was reamed without guide use for both groups, and the guide system was not designed to assist in acetabular orientation.

*Gap volume* was defined by using a thresholding algorithm in segmentation software (Mimics, Materialise Medical Imaging Software Suite, Materialise NV, Leuven, Belgium) with a maximum of 250 Hounsfield units to segment the gap between donor and recipient bone (Fig 5). The resultant gap.stl file was imported into 3D modeling software (3Matic, Materialise Medical Imaging Software Suite, Materialise NV, Leuven, Belgium) to quantify the volume in cubic millimeters (Fig 5). *Femoral neck inclination* was measured by calculating the angle between the axes of femoral neck and femoral diaphysis [34]. Firstly, a best fit sphere encompassing the femoral head was generated by selecting the 3D mesh triangles that made up the femoral head a best fit sphere was generated based on the selected triangle mesh [34]. Similarly, femoral diaphyseal axis was defined as the central axis of a best fit cylinder by selecting the 3D

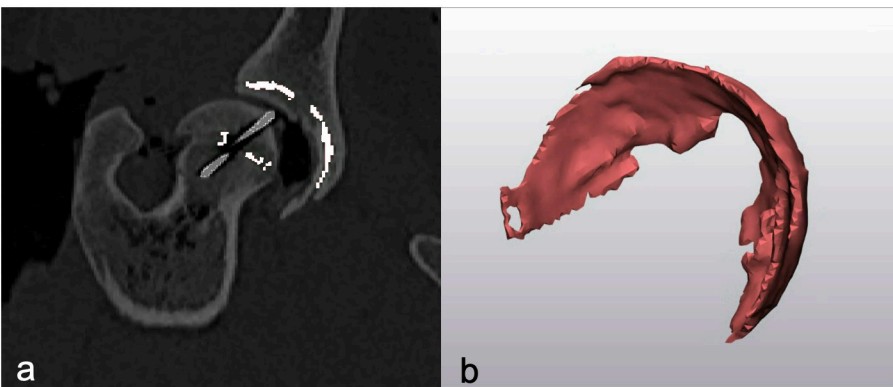

**Fig 5. Segmentation of the gap present between donor and recipient.** (a) Image shows gap between osteochondral allograft and recipient bone. The gap present between donor and recipient femoral head and donor and recipient acetabulum is segmented in white. (b) Image shows resultant 3D rendering of a segmented gap between a donor acetabular osteochondral allograft and the recipient acetabular subchondral bone.

mesh triangles of the diaphysis. A femoral neck axis was defined by a line extending from the center of the femoral head sphere through the center of the femoral neck and intersecting the diaphyseal axis. *Femoral neck length* was obtained by measuring the distance from the center of the femoral head to the intersecting axes of the femoral neck axis and diaphyseal axis [34]. The angle between the femoral neck axis and the diaphyseal axis in the frontal plane represented femoral neck inclination. *Femoral neck version* was determined by measuring the angle created between the femoral neck axis and the transcondylar axis in the transverse plane [34].

Total surgical time, defined as donor and recipient preparation, and gap analysis data were compared between groups. A mirror image of the virtual surgical plan from the custom guide hip was created to provide target parameters for the contralateral side. Femoral neck inclination, version, and length were compared to the virtual surgical plan for both groups and the absolute value difference for each parameter was calculated and compared between groups. All data sets were normally distributed on Shapiro-Wilks test, therefore parameters were compared using a 1-tailed paired t-test. Data is expressed as mean and standard deviation for each parameter. Significance was determined at $p < 0.05$.

## Results

Total surgical time preparing grafts and recipient beds for the guide group was longer than for the freehand group ($p = 0.014$) (Table 1). When compared to free-hand preparation, mean donor femoral preparation time was 2 minutes longer and mean recipient preparation time was 10 minutes longer when using guides ($p = 0.011$ and $p = 0.001$, respectively). No difference in acetabular preparation time was noted between groups. Gap volume at the acetabular and femoral donor-recipient interface was not different between groups. Mean difference between the planned and postoperative version angle was 6.2° lower for the guide group when compared to the freehand group ($p = 0.025$). Mean femoral neck length was 2 mm closer to the plan when using guides than when performing surgery freehand ($p = 0.037$). Accuracy for femoral angle of inclination was not different between groups (Table 1).

## Discussion

This pilot study evaluated the accuracy of bipolar coxofemoral osteochondral allograft transplantation using 3D printed guides and freehand technique in cadaveric dogs. While guides

**Table 1. Data Summary.**

|  |  | Freehand (mean ± SD) | Guide (mean ± SD) | P value |
|---|---|---|---|---|
| **Surgical Time (Min)** | **Donor femur** | 3 ± 1 | 5 ± 0.5 | 0.011* |
|  | **Donor acetabulum** | 16 ± 6 | 15 ± 4 | 0.715 |
|  | **Recipient femur** | 5 ± 2 | 15 ± 4 | 0.001* |
|  | **Total** | 24 ± 7 | 34 ± 6 | 0.014* |
| **Gap (mm³)** | **Femoral** | 131 ± 119 | 114 ± 40 | 0.681 |
|  | **Acetabular** | 1066 ± 327 | 1090 ± 533 | 0.933 |
| **Alignment difference** | **Neck length (mm)** | 3 ± 2 | 1 ± 1.5 | 0.037* |
|  | **Version (degrees)** | 7 ± 4 | 0.8 ± 0.8 | 0.025* |
|  | **Inclination (degrees)** | 8 ± 6 | 4 ± 7 | 0.298 |

Summary of data comparing the mean surgical time, donor-recipient gap, and alignment between groups. Significance denoted by (*).

did not confer an advantage in surgical time, donor-recipient interface gap, or femoral inclination, improved accuracy for femoral version and femoral neck length was achieved when using guides when compared to the freehand technique. Accurate restoration of version and femoral neck length could be clinically relevant, as these parameters are important for hip function and stability, and for success of artificial total hip arthroplasty in dogs [35–37].

We hypothesized that use of our custom guide system would result in faster osteochondral allograft and recipient site preparation compared to freehand techniques; however, freehand preparation was more efficient than using guides. Both increases and reductions in surgical times have been reported with 3D printed custom surgical guides for several orthopedic applications in humans [21, 24, 28, 38]. In our study, we noted that positioning and securing the guides required careful attention to detail to ensure they were applied correctly, which was more time consuming than anticipated. Conversely, the brief visual assessment prior to osteotomy execution translated to a faster procedure. The difference in mean total graft and recipient preparation time in our study was only 10 minutes, which may be clinically acceptable if guides consistently improve accuracy of key aspects of osteochondral allograft transplantation.

The volume of the gap present between donor and recipient subchondral bone was quantified since intimate contact between donor and recipient tissues is recommended for graft stability and eventual osseous integration by creeping substitution [20, 39, 40]. We found no statistically significant differences in femoral and acetabular gap volumes between the two techniques evaluated in this study. In both groups, the mean gap volume between femoral graft and recipient was $\leq 131$ mm$^3$ and there was $\leq 1190$ mm$^3$ for the acetabular graft and recipient. While multiple studies state the importance of accurate graft-recipient fit for biomechanics and osseous union, no published study has quantified minimum donor-recipient gap sizes for appropriate integration to the authors' knowledge [9, 20, 39, 40]. Without an evidence-based benchmark, it is unclear if the gaps noted in the current study would have clinical biomechanical relevance or impact graft incorporation.

Accuracy of restoring femoral version and femoral neck length were superior in the guide group compared to the freehand surgery group. Excessive femoral anteversion following hip replacement in dogs can predispose to patellar and coxofemoral luxation [37]. Malalignment has been cited as a cause of mechanical failure of osteochondral allografts in humans [9]. Proximal femoral malalignment can lead to complications following total hip arthroplasty in dogs including impingement, luxation, and medial patellar luxation [37]. Malalignment of hip prostheses in humans has been shown to precipitate alterations in contact mechanics, increase

polyethylene wear, and predispose to implant failure [36]. In our study, femoral version deviated from the virtual plan by an average of 0.8˚ in the custom guide group and 7˚ in the freehand group. While we cannot assume that the biomechanical properties of a hip osteochondral allograft are the same as a prosthesis, malalignment of osteochondral allografts could potentially have similar long-term detrimental effects. As such, use of custom surgical guides may reduce risk for major complications following bipolar coxofemoral joint osteochondral allograft transplantation compared to use of freehand technique.

The limitations of this study are primarily related to use of cadavers, lack of functional and biomechanical testing, and small sample size. Additionally, dogs used in this study had little or no coxofemoral pathology. Clinically, dogs that would be considered candidates for bipolar coxofemoral osteochondral allograft transplantation are more likely to have anatomical alterations and/or osteoarthritis. Remodeling and osteophytosis can obscure anatomic landmarks, and custom surgical guides may provide greater advantages in clinical cases than our study findings suggest. Greater statistical strength from an increased sample size could uncover trends or significance not appreciated in this study. We did not design and test a guide for the recipient acetabular bed preparation and therefore acetabular orientation was not compared between groups. Lastly, these five cadavers represent our initial experience in bipolar coxofemoral osteochondral allograft in dogs. Increased familiarity with the procedure would likely yield improved outcomes.

In this pilot study using a custom guide system for bipolar coxofemoral osteochondral allograft transplantation, we found that use of guides resulted in longer procedure times; however, guide use also increased surgical accuracy. Although bipolar coxofemoral osteochondral allograft transplantation procedures have not yet been reported clinically in dogs, this study serves as a foundation for developing the technique prior to implementation in clinical cases. Further studies will be required to evaluate the potential benefits of custom guides in dogs with a wider spectrum of anatomic variability and pathology.

## Acknowledgments

We would like to thank Ms. Cat Monger and Mr. Tom DeHaan for their technical support with cadaver surgery as well as the students of the University of Florida's EML 5598 class of Fall 2017 for their engineering contributions.

## Author Contributions

**Conceptualization:** Stanley E. Kim, James L. Cook.

**Data curation:** Christina C. De Armond, Stanley E. Kim.

**Formal analysis:** Christina C. De Armond, Stanley E. Kim.

**Funding acquisition:** Stanley E. Kim.

**Investigation:** Christina C. De Armond, Stanley E. Kim.

**Methodology:** Christina C. De Armond, Stanley E. Kim, Scott A. Banks, Justin D. Keister.

**Project administration:** Christina C. De Armond.

**Resources:** Stanley E. Kim, Adam H. Biedryzcki, Scott A. Banks, Justin D. Keister.

**Software:** Adam H. Biedryzcki, Scott A. Banks, Justin D. Keister.

**Supervision:** Stanley E. Kim, Scott A. Banks, James L. Cook.

**Validation:** Stanley E. Kim.

**Writing – original draft:** Christina C. De Armond, Stanley E. Kim.

**Writing – review & editing:** Christina C. De Armond, Stanley E. Kim, Daniel D. Lewis, Adam H. Biedryzcki, Scott A. Banks, James L. Cook, Justin D. Keister.

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
