## [Decision Letter · Decision Letter 0]

16 Oct 2020

PONE-D-20-20362

Three-dimensional-printed custom guides for bipolar coxofemoral osteochondral allograft in dogs

PLOS ONE

Dear Dr. Kim,

Thank you for submitting your manuscript to PLOS ONE. After careful consideration, we feel that it has merit but does not fully meet PLOS ONE’s publication criteria as it currently stands. Therefore, we invite you to submit a revised version of the manuscript that addresses the points raised during the review process.

We look forward to receiving your revised manuscript.

Kind regards,

Simon Y Tang, PhD

Academic Editor

PLOS ONE

Journal Requirements:

2. Please provide the full name of the IACUC, along with the approval number, in the manuscript.

3. Please clarify in your Methods section the origin of the cadavers and how these were provided.

4.We note that you have stated that you will provide repository information for your data at acceptance. Should your manuscript be accepted for publication, we will hold it until you provide the relevant accession numbers or DOIs necessary to access your data. If you wish to make changes to your Data Availability statement, please describe these changes in your cover letter and we will update your Data Availability statement to reflect the information you provide.

Reviewers' comments:

Reviewer's Responses to Questions

**Comments to the Author**

1. Is the manuscript technically sound, and do the data support the conclusions?

Reviewer #1: Yes

2. Has the statistical analysis been performed appropriately and rigorously? 

Reviewer #1: Yes

3. Have the authors made all data underlying the findings in their manuscript fully available?

Reviewer #1: Yes

4. Is the manuscript presented in an intelligible fashion and written in standard English?

Reviewer #1: Yes

5. Review Comments to the Author

Reviewer #1: PONE-D-20-20362

In this pilot study, authors created and tested a 3D printed surgical guide for osteochondral allograft transplantation in hip joints of dog cadavers. Authors tested a number of relevant parameter but only show moderate success with some negative findings. Despite a number of limitations, the study provides a platform for further investigation in development of custom surgical guide in veterinary medicine for osteochondral allograft transplantation. Overall, this is a very interesting paper and its technical concepts are well-conceived. It is written very well other than a few minor concerns noted below:

Abstract

Abstract is well-written and study design is presented clearly. However, result paragraph needs some details of actual data. For instance, p-values are shown but mean or 95% CI values are not shown. It would be helpful if the data can be included in the abstract.

Introduction

line 88, add “that” after hypothesized…

Methods

Sex of the animals not directly indicated. Age of the animals also need to be added if known.

The sample size is very low, and it would be helpful if a power analysis is shown. This is important since low sample size is certainly one of the reasons as to why results for some important parameters did not reach formal level of significance.

Results

In this section, the actual data is missing. While this reviewer appreciates the citation of results in a Table, it would be helpful to add important numerical data in text, whereas p-values are presented by differences are not.

For all Figures author stated color coding to direct the attention of the reader to a specific location, however this would be problematic on a black/white print. Could author use arrows, circles or similar to point our location of interest. In Fig. 5 pink color needs to be pointed by arrows, although it is clear on color print.

Discussion

Line 26: please replace “initial” with “pilot’

In Discussion, please add a summary paragraph at the end. Currently Discussion ends with limitations, this distracts from the main message.

6. PLOS authors have the option to publish the peer review history of their article (what does this mean?). If published, this will include your full peer review and any attached files.

Reviewer #1: No

---

## [Author Response · Author response to Decision Letter 0]

24 Nov 2020

Dear Editor and Reviewer,

We thank you for your time and constructive comments regarding our submission entitled Three-dimensional-printed custom guides for bipolar coxofemoral osteochondral allograft in dogs. 

Please find our changes below:

Editor Comments:

RESPONSE: Formatting has been corrected to the best of our knowledge. Please let us know if there are further formatting issues.

2. Please provide the full name of the IACUC, along with the approval number, in the manuscript.

RESPONSE: Requested IACUC details have been added to paragraph 1 of the “Materials and methods” section.

3. Please clarify in your Methods section the origin of the cadavers and how these were provided.

RESPONSE: Requested cadaver details have been added to details have been added to paragraph 1 of the “Materials and methods” section.

4.We note that you have stated that you will provide repository information for your data at acceptance. Should your manuscript be accepted for publication, we will hold it until you provide the relevant accession numbers or DOIs necessary to access your data. If you wish to make changes to your Data Availability statement, please describe these changes in your cover letter and we will update your Data Availability statement to reflect the information you provide. 

RESPONSE: DOI has been provided

Reviewer Comments:

Abstract

Abstract is well-written and study design is presented clearly. However, result paragraph needs some details of actual data. For instance, p-values are shown but mean or 95% CI values are not shown. It would be helpful if the data can be included in the abstract.

RESPONSE: The Abstract now contains the means for each of our significant findings. 

Introduction

line 88, add “that” after hypothesized…

RESPONSE: Line 111: “that” has been added following “hypothesized”

Methods

Sex of the animals not directly indicated. Age of the animals also need to be added if known. The sample size is very low, and it would be helpful if a power analysis is shown. This is important since low sample size is certainly one of the reasons as to why results for some important parameters did not reach formal level of significance.

RESPONSE: The methods section now includes the sex of the animals, approximate age (skeletally mature), and our pre-study power analysis.

Results

In this section, the actual data is missing. While this reviewer appreciates the citation of results in a Table, it would be helpful to add important numerical data in text, whereas p-values are presented by differences are not.

RESPONSE: Means of our significant findings are now included in the text of the results section as well as in Table 1.

For all Figures author stated color coding to direct the attention of the reader to a specific location, however this would be problematic on a black/white print. Could author use arrows, circles or similar to point our location of interest. In Fig. 5 pink color needs to be pointed by arrows, although it is clear on color print.

RESPONSE: Figures 1 and 5 and their corresponding figure captions have been amended to account for black/white print

Discussion

Line 26: please replace “initial” with “pilot’

RESPONSE: - Line 399: “Initial” has been changed to “pilot” 

In Discussion, please add a summary paragraph at the end. Currently Discussion ends with limitations, this distracts from the main message.

RESPONSE: - The Conclusion paragraph is now the last paragraph of the Discussion and has a summary sentence.

---

## [Editor Report · Decision Letter 1]

7 Dec 2020

Three-dimensional-printed custom guides for bipolar coxofemoral osteochondral allograft in dogs

PONE-D-20-20362R1

Dear Dr. Kim,

We’re pleased to inform you that your manuscript has been judged scientifically suitable for publication and will be formally accepted for publication once it meets all outstanding technical requirements.

Kind regards,

Simon Yue-Cheong Tang, PhD

Academic Editor

PLOS ONE
---

## [Editor Report · Acceptance letter]

2 Jan 2021

PONE-D-20-20362R1 

Three-dimensional-printed custom guides for bipolar coxofemoral osteochondral allograft in dogs 

Dear Dr. Kim:

I'm pleased to inform you that your manuscript has been deemed suitable for publication in PLOS ONE. Congratulations! Your manuscript is now with our production department. 

Kind regards, 

on behalf of

Dr. Simon Yue-Cheong Tang 

Academic Editor

PLOS ONE